# Fabrication of Ultra-Stable and Customized High-Temperature Speckle Patterns Using Air Plasma Spraying and Flexible Speckle Templates

**DOI:** 10.3390/s23177656

**Published:** 2023-09-04

**Authors:** Ning Lu, Liping Yu, Qianqian Wang, Bing Pan

**Affiliations:** 1National Key Laboratory of Strength and Structural Integrity, School of Aeronautic Science and Engineering, Beihang University, Beijing 100191, China; 2Shandong Institute of Nonmetallic Materials, Jinan 250031, China

**Keywords:** high-temperature speckle patterns, digital image correlation, thermal deformation, speckle fabrication

## Abstract

Reliable and accurate full-field deformation measurements at elevated temperatures using digital image correlation (DIC) require stable and high-contrast high-temperature speckle patterns to be prepared on the sample surface. However, conventional high-temperature speckle patterns fabricated by the existing methods possess several limitations, e.g., easily fail to preserve original pattern features due to the harsh environment and heavily dependent on the operator’s experience. In this study, we propose a reliable and reproducible high-temperature speckle fabrication method based on air plasma spraying (APS) and flexible speckle templates. This method involves covering the sample surface with pre-designed speckle templates and then spraying the melted speckle powders onto the specimen surface using an air plasma spray technique to obtain customized speckle patterns. The validity of the proposed method was verified by the speckle fabrication on both planar and curved samples and heating tests with these samples. Experimental results demonstrate that the speckle patterns made by the proposed method adhere well to the sample surface, remain stable during the heating process, and exhibit excellent agreement with the reference values in terms of the thermal expansion coefficients. The proposed method provides a reliable and efficient way to create customized and stable speckle patterns for accurate high-temperature DIC measurements.

## 1. Introduction

Recent advancements in industries such as nuclear, aerospace, and aero-engine have demanded the materials and structures to be served at more severe and extreme environments throughout the service period, e.g., higher temperatures, faster speeds and heightened stress [1,2]. These harsh conditions could lead to a decline in the mechanical properties of materials as well as structural damages, including significant deformations, cracks, dislocations, or delamination, both on the surface and within the structures [3,4]. It is therefore imperative to characterize the thermo-mechanical response of these materials through high-temperature tests during the design phase. This is crucial for assessing the reliability and stability of both the materials and structures, as well as gaining insight into the underlying mechanisms governing their mechanical behavior. Thus, achieving accurate measurement of surface deformation at elevated temperatures is a key requirement in this endeavor.

Since 2010, digital image correlation (DIC) has become the most practical and powerful experimental tool for high-temperature deformation measurements due to its simplicity of preparation, wide applicability and low experimental requirements [5,6,7]. As an image-based deformation measuring technique, DIC relies on digital image processing and numerical computation. Through the continuous efforts on the key technical issues in high-temperature deformation measurement, DIC techniques have been increasingly applied for high-temperature deformation measurement of various materials and structures [8,9,10]. For example, by suppressing the thermal radiation from the heated sample and heating elements using active imaging technique (i.e., bandpass filter imaging and monochromatic lighting) [11], the applicable temperature range of high-temperature DIC methods has been extended to 2000 °C or even 3000 °C [12,13,14].

To perform accurate high-temperature DIC measurements, the surface of the test specimen usually should be prepared with reliable speckle patterns, which serve as the deformation carrier for DIC analysis. Except for the basic requirements (i.e., high contrast, randomness, isotropy and stability) of preparing the speckle patterns, good speckle patterns for high-temperature measurements should maintain their original feature as possible and adhere tightly to the sample surface without debonding, cracking and peeling off during the heating process. In some cases, the natural texture on the sample surface can provide plenty of intensity variations for shape and deformation measurements [15]. For instance, the blunt cone texture with sufficient image contrast was used as speckle patterns to accomplish 3D shapes and recession measurements of materials in high-temperature wind tunnel environment [16]. However, the surface natural texture may degrade with the heating process due to the chemical reaction (e.g., oxidation), and more importantly, most test samples are short of surface texture or features for the image correlation analyses. In addition, laser speckle generated by illuminating an optically rough surface with coherent light has recently been recommended for high-temperature DIC measurements [17,18], as it offers prominent advantages of simplicity, non-invasive measurement and anti-debonding property at elevated temperatures. Nevertheless, laser speckle is only applicable for 2D deformation measurement, and will encounter severe decorrelation problem when the sample surface undergoes considerable rigid-body translation or deformation [19]. Therefore, despite the non-invasive nature of natural texture and laser speckle, it is necessary to develop a suitable technique to fabricate satisfying high-temperature speckle patterns for the high-temperature DIC measurement.

Many efforts have been devoted to fabricate stable and high-contrast high-temperature speckle pattern for high-temperature deformation measurement. These approaches include but not limited to abrading the sample surface, high-temperature paint spraying, splashing the mixed speckle pattern and plasma spraying. For instance, Grant et al. [20] prepared the speckle patterns by abrading the sample surface with a 1200 grit silicon carbide paper to assure a homogeneous distribution of scratches without any preferential directions. In addition, customizable engraving speckles with femtosecond laser were also proposed for the DIC measurement at elevated temperatures [21]. Although both these two methods do not require additional speckle materials, the specimen surface could be damaged by the femtosecond laser or the silicon carbide paper, leading to the concentration of residual stress [11]. By contrast, high-temperature paint spraying or brushing are mostly adopted for the high-temperature speckle fabrication due to their non-destructive nature and broad applicability [14], which mainly involves depositing a mixture of high-temperature-resistant materials and inorganic high-temperature adhesive onto the surface of the sample. For these approaches, the ceramic oxides (e.g., cobalt oxide (CoO_2_) and zirconium dioxide (ZrO_2_) [22,23,24]) are commonly used as the speckle pattern material because of its stability and heat-resisting property at elevated temperatures [25]. Nevertheless, when exposed to ultra-high-temperature environment or undergo thermo-mechanical coupling loading, the speckles made by high-temperature paint spraying may still experience spalling or peeling-off. To address this problem, an approach based on air plasma spraying was proposed for fabricating high-temperature speckle pattern [26], which was demonstrated at the temperature up to 2600 °C. Note that plasma spraying is capable of melting materials and depositing them onto the surface of the sample, resulting in higher bonding strength. However, in this method, the specimen surface was merely coated with a uniformly distributed wire mesh and then sprayed with speckle materials. As a result, the fabricated speckle patterns exhibited a sparse and periodical spatial distribution, which may degrade the measuring accuracy or even lead to a failure during DIC analysis.

In this work, we propose a simple and repeatable high-temperature speckle patterns fabrication method through air plasma spraying and user-specific speckle templates. The process mainly involves applying a specially designed speckle template onto the specimen surface, and then spraying the speckle powders to generate customized speckle patterns. To validate the effectiveness of this method, both planar and curved specimens were sprayed with speckle patterns and conducted heating tests, respectively. The performance of the proposed speckle patterns was thoroughly examined and the measured results of thermal deformation were compared with the reference values. This approach offers a practical and effective alternative for high-temperature speckle fabrication, which can be applied to high-temperature testing of metallic materials, as well as thermal deformation analysis in aerospace vehicles and other key structures.

## 2. Methods for Fabricating Ultra-Stable and Customized High-Temperature Speckle Patterns

### 2.1. Air Plasma Spray

Among various thermal spraying technologies, APS is a prominent method used for applying thick coatings onto substrate surfaces for protection. Figure 1 illustrates a schematic representation of a typical APS system. The plasma jet is generated by a DC arc formed between the cathode and the nozzle wall, which functions as the anode [27]. The plasma jet then emerges as small plasma columns from the DC plasma torch, where a gas with higher thermal conductivity (e.g., H_2_ or He) is mixed with a gas of high atomic weight (e.g., N_2_ or Ar). Fine particles suspended in the carrier gas are heated and accelerated by the plasma jet, and subsequently sprayed onto the substrate surface. This sudden impact and deceleration between the arriving particles and the surface results in an increased pressure, inducing lateral flow of liquid material. As a consequence, the hot liquid spreads outward from the impact site and solidifies, forming a lamella. To ensure smooth spraying, it is preferable to employ particles with sizes ranging from 10 to 100 μm. Ultimately, the accumulation of these lamellas builds up a stable coating that can vary in thickness from a few tens of micrometers to a few tens of millimeters. Typically, the operating voltage of the plasma torch remains below 70 V, although the working current can reach as high as 1000 A. At the nozzle exit of the plasma torch, the gas temperature may rise to a range of 10,000–12,000 K, and the velocity primarily ranges from 400 to 2600 m/s, enabling the melting of most solid materials [28].

Compared to other spraying techniques, APS technique exhibits several characteristics that make it advantageous for various applications. First, it demonstrates high applicability by utilizing a high gas temperature, allowing for the heating of diverse materials into a molten state for spraying. Second, APS provides excellent controllability, as the deposition efficiency is high and the coating thickness can be adjusted by manipulating the spraying time and velocity distribution. Third, the stability of APS coatings is noteworthy, as they adhere well to the substrate and exhibit a dense structure with a low porosity ranging from 1–10%. Moreover, APS has a minimal heat effect on the substrate, with surface temperatures generally remaining below 600 K during the painting process, thus causing negligible structural changes in the materials. These advantages have made air plasma spray particularly valuable in structural repair and protection applications, including the fabrication of thermal barrier coatings in gas turbines to provide thermal insulation.

### 2.2. High-Temperature Speckle Pattern Fabrication Using Air Plasma Spray and Flexible Speckle Templates

The implementation process of the proposed APS-based speckle pattern fabrication was shown in Figure 2, which mainly comprises three steps:

Step 1: speckle pattern design. Digital speckle pattern (DSP), characterized by randomly positioned speckles, is widely utilized for the error analysis in DIC because of its simpleness and controllable randomness [29,30]. The pattern can be generated through computer simulations by adjusting multiple parameters such as speckle size *D*, speckle density *ρ*, and speckle variation *δ* [31]. The generation process involves creating a grid with a step size of *L* = *D*/*ρ* based on the specified parameters, followed by the placement of speckles on the grid points. Then, each speckle is assigned a unique displacement (*d_x_*, *d_y_*), where both displacement components vary randomly within the range of [−*δL*/2, *δL*/2]. By adjusting these parameters, researchers can control the spatial distribution of speckles, allowing for customization and optimization according to specific experimental requirements. Su et al. [31,32] undertook an extensive inquiry into the influence of these parameters on the accuracy of DIC measurements, and confirmed that a speckle size of 4 pixels was optimal for analysis, aligning with previous research findings [33]. However, replicating DSP into actual speckles, especially for high-temperature deformation measurements, poses inherent challenges. The process of duplicating high-temperature speckle patterns is intricate. Moreover, the simulated speckle pattern may exhibit numerous closely arranged tiny speckles, particularly for a small field-of-view (FOV), which could lead to localized overheating, deformation, or even damage during subsequent mechanical processing and laser drilling for speckle template. Lecompte et al. [34] have identified that evenly and randomly distributed speckle pattern was sufficient for accurate deformation measurement. Therefore, the pursuit of the smallest possible speckles in practical applications is not necessary. To ensure the replicability of the proposed methodology in this study, it is recommended to modify or expand the speckle size as needed during the preparation of actual high-temperature speckle patterns. Based on the spatial resolution of the experimental system, which is determined by the camera resolution *w_camera_* × *h_camera_* (in unit of pixels) and the FOV *W_system_* × *H_system_* (in unit of millimeters), the corresponding speckle diameter *D* in millimeters can be obtained as follows:(1)$$D=n\times \frac{W_{system}\times H_{system}}{w_{camera}\times h_{camera}}$$
where *n* denotes the predetermined pixel size of speckle. Aside from the speckle size, the remaining parameters of speckle pattern are determined in accordance with the recommended values provided in the article [31]. After introducing these into open-source or commercial software (e.g., Glare), a simulated speckle pattern was generated.

Step 2: speckle template fabrication. The speckle pattern generated in Step 1 is then imported into the three-dimensional modeling software, such as SolidWorks, in the form of an image. Subsequently, the characteristics of speckle distribution are extracted from the imported image through features recognition to create a sketch. The sketch should be strategically modified to incorporate auxiliary appendages that correspond to the shape and contours of the speckle pattern, ensuring a close fit between template and specimen. If necessary, the sketch can be further edited by adding or removing entities, adjusting dimensions, or modifying constraints. Once the modifications are complete, the sketch is exported in either DWG or DXF format. Note that during the plasma spraying process, the substrate surface can reach temperatures as high as 400 K. therefore, it is advisable to choose materials with elevated melting points for the template preparation. Stainless steel is highly recommended due to its exceptional mechanical property and excellent heat resistance. Moreover, the use of stainless steel enables the possibility of reusing the speckle template, contributing to its cost-effective and sustainable application. Following selection, the raw material can be processed with mechanical machining and laser drilling according to the provided file, resulting in a customized speckle template. It is noteworthy that maintaining a thinner template thickness offers greater flexibility, allowing for easier installation. Therefore, it is advisable to maintain the template thickness below 0.5 mm.

Step 3: APS spraying. Before spraying, compressed air is commonly used to perform sandblasting on substrates, employing materials such as corundum, quartz, and iron sand to eliminate impurities, enhance surface roughness, and improve adhesion of the substrate. Subsequently, heat-resistant paints are applied as a primer to the substrate surface using a plasma torch. Once the primer is in place, the customized speckle template is positioned on the specimen surface, and securely fastened using auxiliary handles. Then, speckle powders are deposited onto the substrate using APS technique. It is crucial to ensure that there exists ample color contrast and no chemical reaction between the primer and speckle constituents. For instance, white zirconia powders can be utilized to form speckle patterns, while black silicon carbide powders can be employed for primer painting. In case where the test sample exhibits significant color contrast with the speckle powders, pretreating the sample by primer spraying is unnecessary. Ultimately, this process yields customized high-temperature speckle patterns onto the specimen surface.

## 3. Validations of APS-Based High-Temperature Speckle Patterns

### 3.1. Fabrication and Characterization of APS-Based Speckle Patterns

During the spraying process, the molten speckle powders exclusively traverse the meticulously distributed apertures of the speckle template and then adhere onto the specimen surface. However, it should be noted that excessively diminutive apertures would impede the passage of speckle particles, as the size of particles is usually controlled within the range of 10 to 100 μm. In addition, the practical drilling of micro-pores (with dimensions below 1 mm) using pulsed laser is typically limited by processing capacity and accuracy. Unnecessary machining costs will be increased if the required speckle size is too small. Therefore, it is significant to determine the proper speckle size (in units of millimeter) that meets the dimensional requirements applicable for DIC while ensuring a smooth processing. In this study, the minimum hole size of laser drilling is determined to be 0.3 mm given that the particle size.

To investigate suitable speckle size and APS spraying outcome, several square samples of Nickel-base alloy GH4169 measuring 50 × 50 × 2 mm were employed, as shown in Figure 3a. Taking the speckle pattern simulated with the speckle diameter *D* of 0.5 mm as an example to show the implementation process for spraying APS-based speckle patterns on specimen, as depicted in Figure 3b. Subsequently, a stainless-steel speckle template with a thickness of 0.5 mm was custom-made, replicating the simulated pattern distribution. The template was then affixed to the sample surface using four auxiliary handles, as shown in Figure 3c. Since the blasted sample surface shows black, white 8YSZ powders (8 wt% Y_2_O_3_ Stabilized ZrO) are used to prepare customized speckle patterns. The speckle coating, with a thickness of 70 μm, is then deposited on the GH4169 sample surface using a commercial plasma spray system (BJXT-F4, Beijing Xinte, Beijing, China). The specific APS parameters utilized in this study are provided in Table 1. As demonstrated in Figure 3d, the obtained speckle pattern closely resembled the input speckle pattern.

A similar process for speckle preparation was conducted on other square samples, resulting in the fabrication of speckle patterns with varying speckle diameters (0.3 mm, 0.5 mm, 0.7 mm, 0.9 mm, and 1.1 mm) on their surfaces. Note that when the diameter *D* was set to 0.3 mm, the small and densely distributed speckle holes caused excessive heat generation during the fabrication process of the speckle template, leading to localized deformation. Consequently, it became unattainable to achieve a customized speckle template with the desired distribution, thereby rendering subsequent APS spraying infeasible. By contrast, when the speckle diameter *D* is set to 0.5 mm or larger, the fabrication of the speckle template and speckle spraying proceed smoothly. The obtained speckle patterns exhibit conspicuous appearances and faithfully replicate the imported speckle patterns, showing a close similarity, as illustrated in Figure 4. Furthermore, a supplementary experiment was conducted to assess the feasibility of preparing speckle pattern with speckle diameter *D* of 0.4 mm, but similar localized overheating issues were encountered during the fabrication of template. Consequently, adopting a speckle diameter *D* of 0.5 mm or larger provides better visibility and facilitates the fabrication of the speckle template, enabling the fabrication of reproducible APS-based speckle patterns.

Actually, not all speckle patterns are suitable for conducting DIC analysis when considering a specific FOV. This is due to the fact that speckle patterns created with different speckle diameters can yield distinct histogram distributions, affecting the accuracy of DIC measurement. In order to achieve accurate measurements, it is crucial to evaluate and select an optimal speckle pattern before experiments. To tackle that, Pan et al. proposed a global parameter called the mean intensity gradient (MIG) *δ_f_* by refining the local parameter SSSIG [35].
(2)$$\delta_{f}=\frac{\sum_{i=1}^{w}\sum_{j=1}^{h}\left|\nabla f(x_{ij})\right|}{w\times h}$$
where *w* and *h* (in unit of pixels) are image width and height, $\left|\nabla f(x_{ij})\right|=\sqrt{f_{x}{\left(x_{ij}\right)}^{2}+f_{y}{\left(x_{ij}\right)}^{2}}$ represents the magnitude of the intensity gradient vector at each pixel (*x_ij_*). *f_x_*(*x_ij_*) and *f_y_*(*x_ij_*) are the *x* and *y* directional intensity derivatives, respectively, which can be determined by the gradient operator.

Experiments demonstrated that a speckle pattern with a higher MIG resulted in a smaller bias error and standard deviation error. Despite efforts to fabricate high-temperature speckle pattern, the complex heating environment has made it challenging to achieve a high MIG value *δ_f_*, resulting in a comparatively lower value compared to conventional speckle pattern. In the case of the above sprayed specimens, it is apparent that image with a smaller speckle diameter tend to exhibit higher MIG when captured using a 1440 × 1080 camera sensor with the same exposure time, as shown in Figure 5. In this experimental setup, 1 pixel corresponded to approximately 50 μm, and the speckle diameter *D* value of 0.5 mm was the optimal choice. Moreover, it is advisable to adjust the bore diameter of the speckle template according to the spatial resolution of the experimental system and the MIG values of the captured images, particularly when work with different FOV and imaging systems. As such, researchers can achieve distinguishable speckle patterns that are appropriate for high-temperature deformation measurements.

### 3.2. Thermal Deformation Measurement of a Planar Sample with APS-Based Speckle Pattern

To validate the practicality and accuracy of the proposed APS-based speckle patterns under high-temperature conditions, a heating experiment was conducted on the mentioned GH4169 sample, which was prepared with the speckles of diameter measuring 0.5 mm. The GH4169 alloy, renowned for its exceptional fatigue strength, corrosion resistance, and mechanical performance at elevated temperatures, finds extensive application in the field of aerospace engineering. Detailed information regarding its coefficients of thermal expansion (CTEs) can be found in relevant literature [36]. In addition, its chemical composition is shown in Table 2.

As depicted in Figure 6a, the GH4169 sample was heated by two torch burners, while the images of sample surface were captured using a CCD camera (ME2L-161-61U3M-L, Beijing Daheng Image Vision Co., Ltd., (Beijing, China) sensors:1/2.9″, 1440 × 1080 pixels) equipped with an imaging lens (Xenoplan 1.4/23-0902, Schneider Optics, Inc., (Hauppauge, NY, USA) Bad Kreuznach, Germany). Active imaging devices, including optical bandpass filter (center wavelength: 480 nm; full-width at half-maximum (FWHM) value: 10 nm) and monochromatic light source (480 ± 10 nm, 25 W) were implemented to suppress the thermal radiation from heated object. Note that the speckle diameter *D* of 0.5 mm corresponded to approximately 10 pixels for this experiment setup. The capturing system placed 0.5 m way from the specimen, with its optical axis approximately perpendicular to the nominal sample surface. By adjusting the imaging lens, a clear reference image with good contrast was recorded as shown in Figure 6b. Then, the sample was heated from room temperature to 700 °C, and the infrared thermometer real-time monitored the variations of sample temperature, which is documented as Stage 1. Seven images of the deformed specimen were consecutively recorded at temperatures from 100 to 700 °C with a temperature increment of 100 °C at each step. Subsequently, the torch burners were positioned in front of the speckled surface, enabling a high-velocity airflow with elevated temperature to thoroughly flush the surface, constituting Stage 2. Multiple images of the specimen were recorded at various intervals during the flushing process.

The recorded surface images were analyzed by the self-developed DIC algorithm, for obtaining the full-filed thermal deformation during the stage 1, which directly compares each deformed images captured at different temperatures and the reference image captured at room temperature. As shown in Figure 6b, a rectangular area in the center of sample is defined as the region of interest (ROI). The displacements are calculated at a grid of 87 × 81 points (corresponding to the area of 21.0 × 17.5 mm) with the subset size of 33 × 33 pixels and the grid step of 10 pixels. Figure 7 presents the measured full-field displacements of the GH4169 sample at 700 °C. The *u*− and *v*− displacement after removing the in-plane rigid body motion are shown in Figure 7a and Figure 7b, respectively. The equidistant contour lines, nearly parallel to the *x* and *y* axes, indicate homogeneous thermal expansion occurring in both the horizontal and vertical directions. The radial displacement of test sample is shown in Figure 7c, demonstrating near-uniform thermal expansion, with deformation magnitudes gradually increasing from the center to the edge. The mean values of the strain fields within ROI at 700 °C, in the horizontal and vertical directions, are 10,736 με and 10,779 με, respectively, representing the thermal strains of sample.

After that, radial displacements of the sample at different temperatures are shown in Figure 8, visualized using the same colorbar range. Apparently, approximately concentric circles can be observed, with the magnitude of displacements increasing as the temperature up. The corresponding *x*-directional and *y*-directional thermal strains of the sample at elevated temperatures are provided in Table 3. It is apparent from Table 3 that the measured thermal strains increase with the applied temperatures, and are almost identical in horizontal and vertical directions at each temperature step. This observation is consistent with the isotropic characteristics of GH4169, i.e., the coefficients of thermal expansion (CTEs) along the *x* and *y* directions are equivalent. The experimental measurements of the CTEs exhibit a discrepancy of less than 4% compared to the theoretical values, which can be attributed to the potential variations in the composition of material.

Moreover, Figure 9 shows the surface images of the GH4169 specimen recorded during the Stage 2. It can be observed that as the flushing time increases, the speckle patterns remain tightly adhered to the sample’s surface without significant changes, and the captured images exhibit relatively stable gray intensity. To gain further insights into the adhesion effectiveness of the APS-based speckle patterns, the grayscale distribution of these images is extracted as shown in Figure 10a, demonstrating consistent grayscale distribution curves at different flushing intervals. Furthermore, the characteristic variations of high-temperature speckles during the heating process serve as another crucial factor for assessing speckle quality. As depicted in Figure 10b, the speckle patterns maintain stability with a constant count of 1611 throughout the heating and flushing process, exhibiting no cracks or spalling. With the temperature increase, the MIG values of these images fluctuate around 6.459, with a maximum variation of approximately 0.130. All these results affirm that the proposed speckle patterns closely adhere to the sample’s surface throughout the entire process, confirming its applicability for accurate high-temperature DIC measurements.

## 4. Application to Thermal Deformation of a Turbine Blade under Induction Heating

To demonstrate the applicability of the proposed method for a curved surface, APS-based speckle patterns were sprayed onto a real turbine blade surface (made of DZ5) measuring 52 × 94 mm for deformation measurements at elevated temperatures. The chemical composition of DZ5 is shown in Table 4. It should be mentioned that the speckle template must be thin enough to ensure a stable immobilization onto this type of curved specimen surface. A template thickness of 0.1 mm was employed to achieve a close fit to the blade surface during the spraying process. As depicted in Figure 11a, speckle patterns with speckle diameter of 0.5 mm were applied onto the specimen, which corresponded to approximately 27 pixels in this experiment. In addition, it is also recommended to utilize computer-aided design and precise 3D-printing technology to create templates with same surface curvature as the curved specimen. As shown in Figure 11b, the sprayed blade sample was heated by electromagnetic induction using an induction heating system (SPG-30B, Sichuan dexkcyq Instrument Co., Ltd., Chengdu, China). Eddy currents were generated within the blade specimen and resistance lead to Joule heating, allowing for precise control of the specimen’s temperature, even at elevated levels, in an open-air environment. The temperature of the blade surface was monitored and regulated using an infrared temperature measurement device integrated with the heating system. The established bi-prism-based single-bilateral-telecentric-camera (SBTC) stereo-DIC system [37] included a CCD camera (TXG50, Baumer, Electric AG, sensors:2/3″, 2448 × 2050 pixels), and a bilateral telecentric lens (Xenoplan 1:5, Schneider Optics, Inc., initial working distance: 269 ± 75 mm) equipped with an optical bandpass filter (center wavelength: 480 nm; FWHM value: 20 nm). To accomplish 3D-DIC measurements, a bi-prism (bi-prism angle: 5°, index of refraction of the monochromatic blue light: 1.52537, index of refraction of the natural light: 1.51680) was fixed in front of the lens. At first, a blue light source (480 ± 10 nm, 25 W) was used to illuminate the specimen surface and the reference image of the blade specimen was shown in Figure 11c. Then, the temperature was raised from room temperature to 800 °C in almost 185 s with a temperature increment of 4.5 °C/s, and the specimen surface was recorded at 2 fps simultaneously. A total of 370 images were acquired and the same test was repeated for two times.

Figure 12 displays a series of images of the blade specimen recorded at different temperatures, revealing distinct speckle patterns and consistent gray intensity as the temperature rise. Throughout the heating process, the high-temperature speckle sprayed onto the specimen surface remains tightly adhered, without any significant detachment or damage observed. These findings indicate that the method proposed in this study is also applicable for the fabrication of a high-temperature speckle pattern onto the curved specimen surfaces.

After all the reference and deformation images were recorded, a rectangular area near the center of heating was selected as the ROI (67 × 106 points corresponding to the area of 12.2 × 18.8 mm), as shown in Figure 11c. A 57 × 57 pixels subset size and a 11 pixels grid step were set as the calculation parameter for each point of interest. Then, the displacement and strain maps within the ROIs were obtained using the pointwise least squares fitting algorithm [38,39]. Figure 13 displays the measured full-field displacements of the blade sample at 182 s and a temperature of approximately 800 °C. The *u* − and *v* − displacement without in-plane rigid body motion are shown in Figure 13a and Figure 13b, respectively. The uniformly distributed contour lines nearly parallel with the y and x axes, demonstrating that homogeneous thermal expansion occurred in both horizontal and vertical directions. Furthermore, the radial displacements exhibit an approximately concentric circular pattern, as shown in Figure 13c. The average strains of the sample strain maps are extracted as the measured strains. However, due to minor curvature on the blade surface, there is a slight difference in the mean strains between the horizontal and vertical directions, measuring at 11,283 με and 11,636 με, respectively. Moreover, all of these recorded images are analyzed and the measured strains are plotted as a function of the temperature, as shown in Figure 14. It should be noted that the strain measurement results in the range of 25–200 °C are omitted since the infrared radiation system’s measuring range exceeds 200 °C. The agreement between the measurement results and reference values [40] confirms the feasibility and validity of the proposed speckle patterns for high-temperature deformation measurements of curved specimens.

## 5. Conclusions

This work proposes a customized speckle fabrication method based on the APS technique and flexible speckle templates for high-temperature DIC measurements. With this technique, the speckle powders are melted and sprayed onto the specimen surface through the speckle template to fabricate the customized speckle pattern, which aligns with the desired speckle distribution. The speckle size should be adjusted according to the spatial resolution of the experimental system and the MIG values of the captured images, ensuring the best performance of the DIC-based measurements. The validity of the method was confirmed through the fabrication of speckle patterns on both planar and curved samples, as well as the subsequent heating tests conducted on these samples. Experimental results show that the speckle patterns adhere tightly to the sample surface during heating, and the measured results exhibit good agreement with the reference values in the handbook [36,40].

In conclusion, the proposed approach offers a promising solution for creating ultra-stable and customized high-temperature speckle patterns, which provides new avenues for performing deformation measurements at even harsher scenarios (e.g., high-temperature wind tunnel), and demonstrates great potential in thermo-mechanical coupling tests (e.g., elevated temperature tensile test). By enabling an accurate and reliable measurement of deformations under these conditions, this approach can help open up new possibilities for studying and improving the performance of materials and structures subjected to extreme thermal environments.

## Figures and Tables

**Figure 1 sensors-23-07656-f001:**
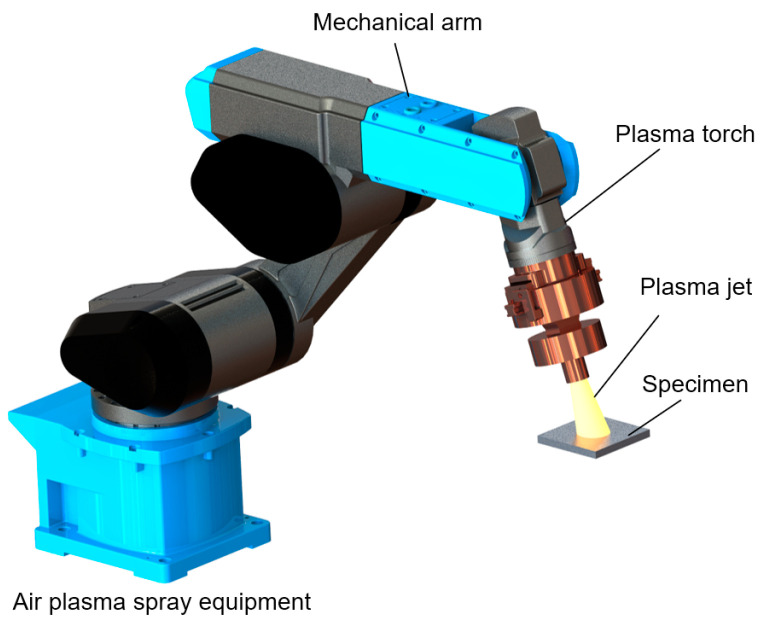
The schematic diagram of general APS system.

**Figure 2 sensors-23-07656-f002:**
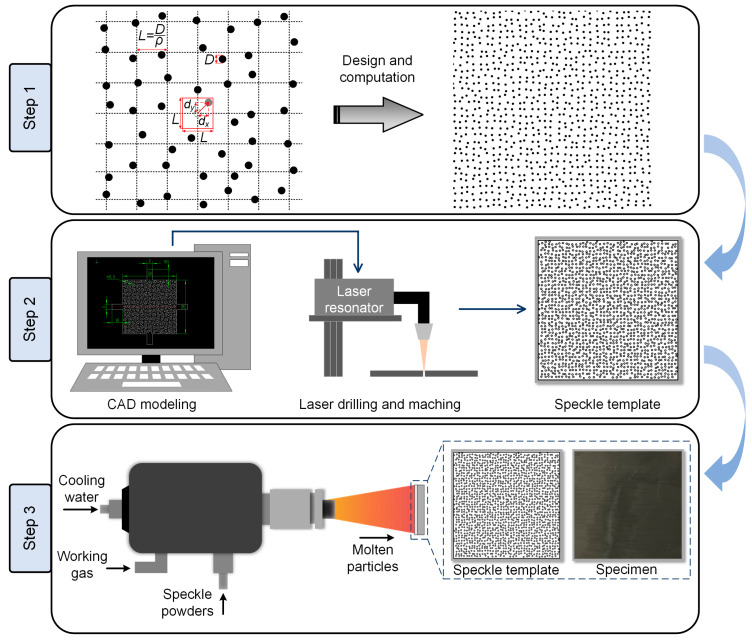
The main procedure for fabricating speckle patterns by using APS and speckle template.

**Figure 3 sensors-23-07656-f003:**
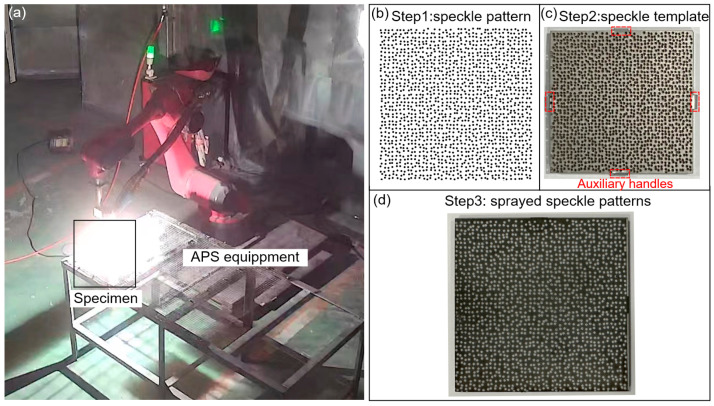
The procedure of speckle fabrication for GH4169 sample: (**a**) spraying process, (**b**) the speckle pattern, (**c**) the speckle template installed on the sample surface and (**d**) the ultimate speckle pattern obtained via the APS technique.

**Figure 4 sensors-23-07656-f004:**
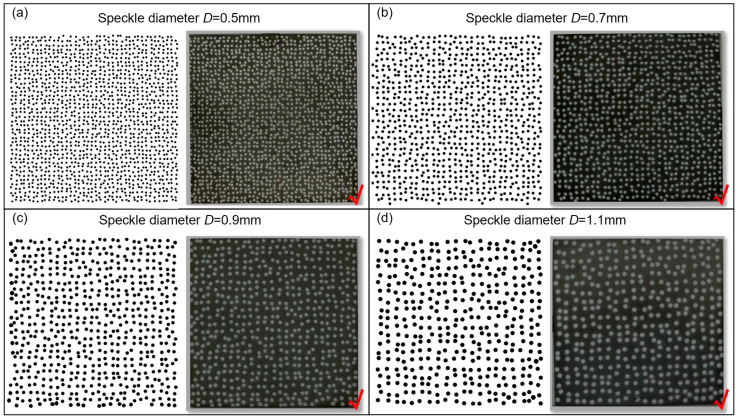
The speckle patterns fabricated on the surface of square GH4169 samples with different speckle diameter *D*: (**a**) 0.5 mm, (**b**) 0.7 mm, (**c**) 0.9 mm and (**d**) 1.1 mm.

**Figure 5 sensors-23-07656-f005:**
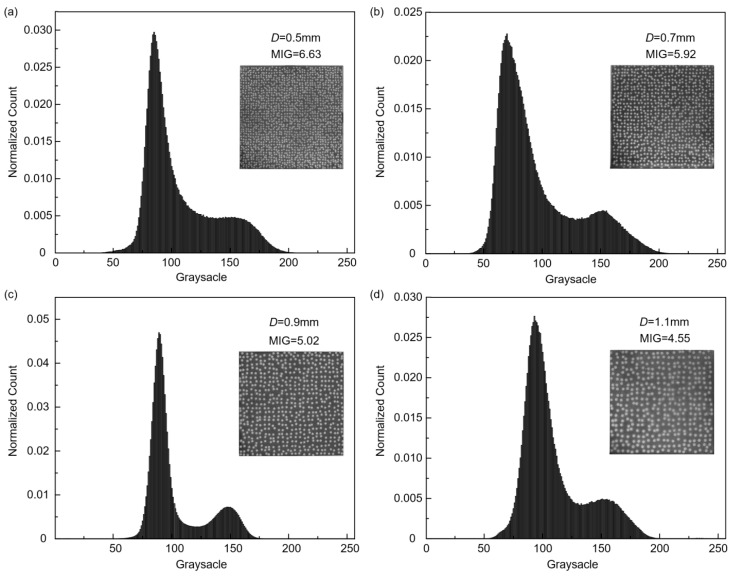
The grayscale distribution and MIG value of speckle pattern with different speckle diameter *D*: (**a**) 0.5 mm, (**b**) 0.7 mm, (**c**) 0.9 mm and (**d**) 1.1 mm.

**Figure 6 sensors-23-07656-f006:**
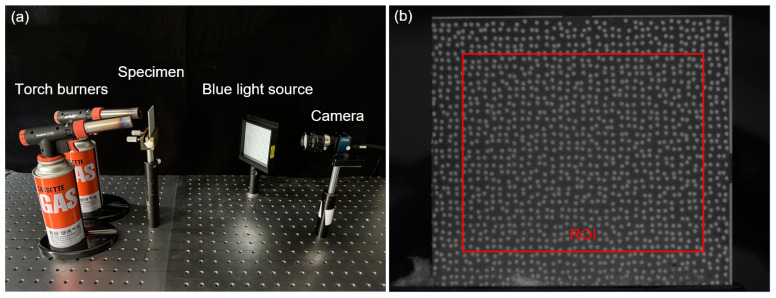
(**a**) The experimental arrangements for measuring the thermal deformation and (**b**) the digital image recorded at the room temperature.

**Figure 7 sensors-23-07656-f007:**
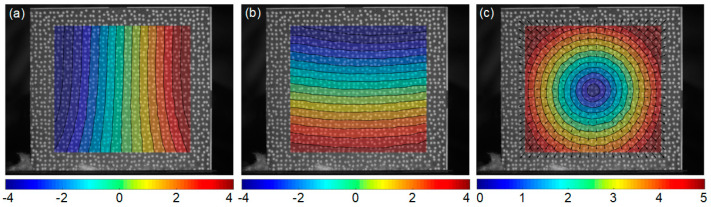
Full-field thermal deformation of the GH4169 sample at 700 °C measured by experimental system: (**a**) *u* − displacement, (**b**) *v* − displacement and (**c**) resultant displacement (units: pixel).

**Figure 8 sensors-23-07656-f008:**
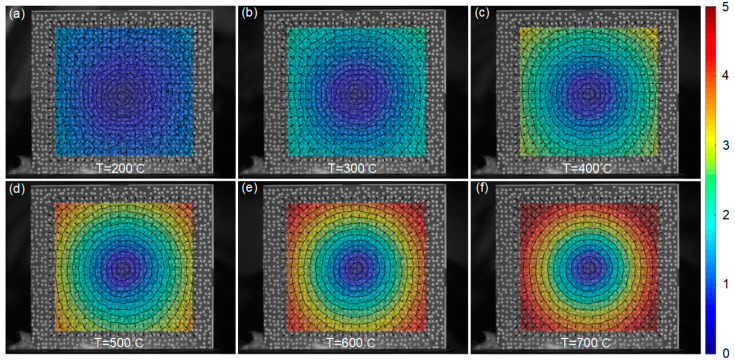
Radial displacements of the GH4169 sample at various temperatures measured by experimental system: (**a**) 200 °C, (**b**) 300 °C, (**c**) 400 °C, (**d**) 500 °C, (**e**) 600 °C and (**f**) 700 °C (units: pixel).

**Figure 9 sensors-23-07656-f009:**
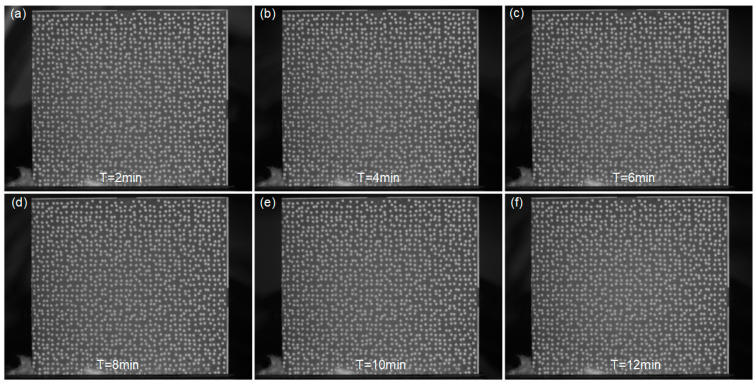
Surface images of the GH4169 sample captured after flushing for (**a**) 2 min, (**b**) 4 min, (**c**) 6 min, (**d**) 8 min, (**e**) 10 min and (**f**) 12 min.

**Figure 10 sensors-23-07656-f010:**
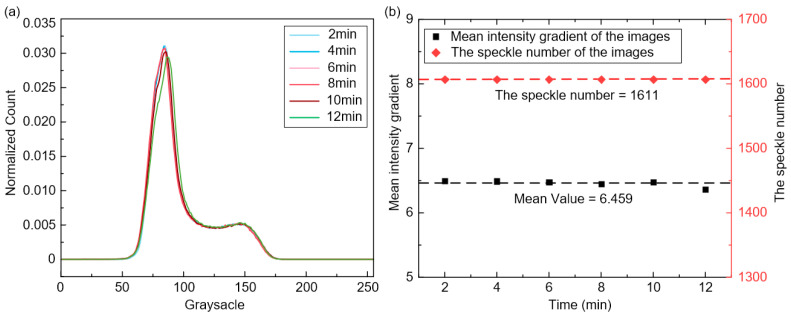
Characteristic changes of speckle patterns within ROI of the recorded images at different intervals: (**a**) grayscale distributions, (**b**) the speckle number and MIG values.

**Figure 11 sensors-23-07656-f011:**
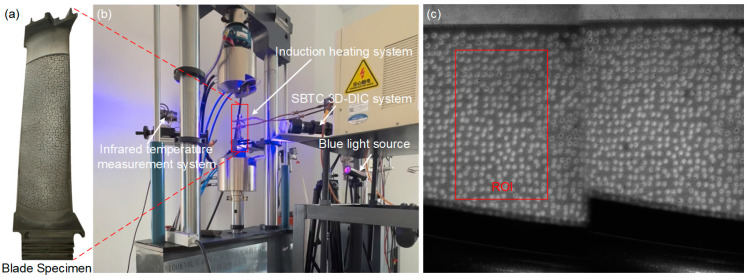
(**a**) The sprayed blade specimen, (**b**) experimental arrangements for measuring the thermal deformation and (**c**) the reference image of the specimen captured at room temperature.

**Figure 12 sensors-23-07656-f012:**
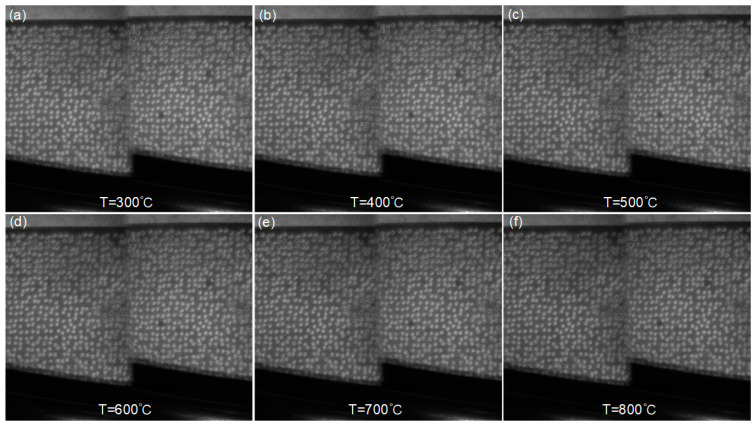
Surface images of the curved blade sample captured by the experimental system at temperatures of (**a**) 300 °C, (**b**) 400 °C, (**c**) 500 °C, (**d**) 600 °C, (**e**) 700 °C and (**f**) 800 °C.

**Figure 13 sensors-23-07656-f013:**
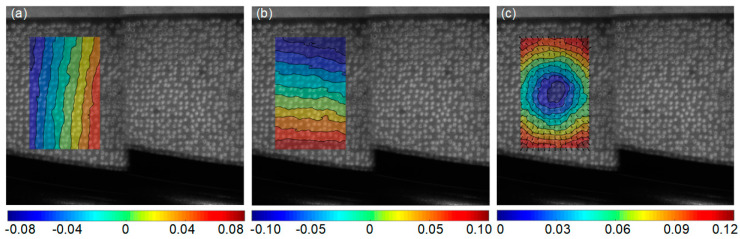
Full-field thermal deformation of the blade sample at 182 s, at a temperature of approximately 800 °C: (**a**) *u*−displacement, (**b**) *v*−displacement and (**c**) resultant displacement (units: mm).

**Figure 14 sensors-23-07656-f014:**
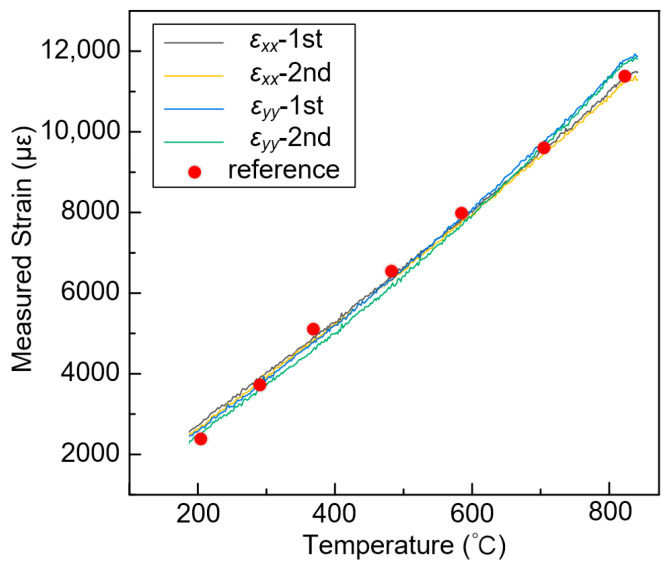
The comparison of measured thermal strains (color lines) and published data [40] (red dots) as a function of the measured temperature on the blade specimen surface.

**Table 1 sensors-23-07656-t001:** APS parameters used for spraying speckle patterns.

Parameter	Unit	Value
Plasma arc current	A	620
Plasma arc voltage	V	75
Flow rate of primary gas (Ar)	L/min	55
Flow rate of secondary gas (H_2_/N_2_)	L/min	8
Flow rate of speckle powders	g/min	40
Spray distance	mm	85
Spray speed	m/s	350
Torch traverse speed	mm/s	500

**Table 2 sensors-23-07656-t002:** Chemical composition of GH4169 (mass fraction, %).

Ni	Cr	C	Mn	Ca	S	P	Mo
50.0~55.0	17.0~21.0	0.02~0.06	≤0.35	≤0.01	≤0.35	≤0.015	2.80~3.30
Al	Ti	B	Nb	Co	Cu	Fe	
0.30~0.70	0.75~1.15	≤0.006	5.00~5.50	2.80~3.30	≤0.30	Bal	

**Table 3 sensors-23-07656-t003:** Measured results of the GH4169 sample at various temperatures.

Temperature/°C	Reference Value of CTE [36]	*ε_xx_*/με	Measured CTE	Relative Error	*ε_yy_*/με	Measured CTE	Relative Error
200	13.0	2533	12.66	2.56%	2568	12.84	1.23%
300	13.5	4127	13.75	1.92%	4194	13.98	3.58%
400	14.1	5733	14.33	1.65%	5819	14.54	3.19%
500	14.4	7221	14.44	0.29%	7271	14.54	0.98%
600	14.8	8917	14.86	0.42%	8971	14.95	1.02%
700	15.4	10,736	15.33	0.41%	10,779	15.38	0.01%

**Table 4 sensors-23-07656-t004:** Chemical composition of DZ5 (mass fraction, %).

C	Cr	Ni	Co	W	Mo	Al	Ti
0.07~0.15	9.5~11.0	Bal	9.5~10.5	4.5~5.5	3.5~4.2	5.0~6.0	2.0~3.0
B	Zr	Mn	Si	P	S		
0.01~0.02	≤0.10	≤0.50	≤0.50	≤0.02	≤0.01		

## Data Availability

The data presented in this study are available upon request from the corresponding author.

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
