# Peer review of "Fabrication of Ultra-Stable and Customized High-Temperature Speckle Patterns Using Air Plasma Spraying and Flexible Speckle Templates"

_sensors, 2023, doi:10.3390/s23177656_

Round 1

Reviewer 1 Report

This manuscript proposes a customized speckle fabrication method based on the APS technique for high-temperature DIC measurements. Although this manuscript lacks theoretical innovation, it has important practical application significance. Therefore, the reviewer suggests that this manuscript can be accepted after minor revisions.

My comments are as follows:

1.     The experimental part only shows the results at 800 , which does not show a great contribution in temperature compared with Ref.27. It is recommended to conduct experiments in a higher temperature environment and obtain the highest temperature at which the proposed method can work.

2.     How the reference values in Figure 14 were obtained

3.     In the introduction, the author summarized various speckle making methods, but did not compare them with the proposed method in the experimental section. The reviewer believes that adding comparative experiments will better reflect the importance of this paper.

There are no obvious issues with English writing,

Reviewer 2 Report

In present paper, the authors propose a reliable and reproducible high-temperature speckle fabrication method based on air plasma spraying (APS) and flexible speckle templates. This method involves covering the sample surface with pre-designed speckle templates and then spraying the melted speckle powders onto the specimen surface using an air plasma spray technique to obtain customized speckle patterns. Some issues should be adressed:

(1) Page 2, Line 90, "......(CoO2) and zirconium dioxide (ZrO2) [22–24]) are commonly used as the speckle pattern......", the "2" in "CoO2 and ZrO2" should be subscript format;

(2) Page 10, Line 353, the table should be  numbered as "Table 2";

(3) Page 10, Line 342-345, "It is evident that the thermal 342 strains increase with higher temperatures, and the resulting strains in horizontal and vertical directions are almost identical. This finding aligns with the isotropic nature of the GH4169 used in this study, which exhibits equal CTEs in the x and y directions", the authors are suggested to explain it more detaily;

the paper is well organised and can be accepted after minor reversion.

Reviewer 3 Report

The manuscript “Fabrication of ultra-stable and customized high-temperature speckle patterns using air plasma spraying and flexible speckle templates” addresses an actual problem related to application of DIC to high-temperature strain measurements. For solving the problem, a stable and high-contrast speckle patterns should be applied to the sample’s surface. An air plasma spraying (APS) and flexible speckle templates are employed for speckle deposition. The validity of the proposed method was verified on both planar and curved samples. It was shown that the speckle patterns adhered well to the sample surface and kept stable during the heating. The excellent agreement with the reference values was attained when measuring thermal expansion coefficients. It is concluded that the developed method ensures reliable and efficient way for accurate high-temperature DIC measurements.

 The state of the art is perfectly characterized.

The experimental procedure is explained in due details. The experiments might be reproduced elsewhere.

The experimental results are clearly described and explained.

The Conclusion summarizes the outcomes and shows the prospects.

The manuscript falls within the scope of the journal of Sensors.

The level of English language is OK.

 The manuscript might be accepted for the publication.

In order to perfect the manuscript, the following issues are to be addressed by the authors.

 1. Page 4. “APS has a minimal heat effect on the substrate, with surface temperatures generally remaining below 600K during the painting process”. Did you measure the temperature of the substrate? In addition, the latter is usually artificially heated up in order to improve the adhesion.

2. Page 5. “Note that during the plasma spraying process, the substrate surface can reach temperatures as high as 400K”. This is too low for plasma spraying.

3. Page 6. The chemical composition of Nickel-base alloy GH4169 is to be given.

4. Sprays speed was 350 m/s (page 7, table 1) while the template thickness was just 0.5 mm. How the bending or damaging of the template was avoided under erosive action of hard zirconia particles?

5. Page 9. “The recorded surface images were analyzed by regular DIC algorithm”. Was it a commercial software or homemade one? Please provide the details.

6. Page 12. What is DZ5? The non-Chines readers are not familiar with this alloy. The chemical composition is to be added.

7. Page. 14. It is recommended to mention in the conclusion that optimum speckle parameters were established to ensure the best performance of the DIC-based measurements.

Reviewer 4 Report

Please find the attached PDF file containing comments and suggestions for the authors.
